# How Asian Breast Cancer Patients Experience Unequal Incidence of Chemotherapy Side Effects: A Look at Ethnic Disparities in Febrile Neutropenia Rates

**DOI:** 10.3390/cancers15143590

**Published:** 2023-07-12

**Authors:** Zi Lin Lim, Peh Joo Ho, Mikael Hartman, Ern Yu Tan, Nur Khaliesah Binte Mohamed Riza, Elaine Hsuen Lim, Phyu Nitar, Fuh Yong Wong, Jingmei Li

**Affiliations:** 1Genome Institute of Singapore (GIS), Agency for Science, Technology and Research (A*STAR), 60 Biopolis Street, Genome, Singapore 138672, Singapore; lim_zi_lin@gis.a-star.edu.sg (Z.L.L.); ho_peh_joo@gis.a-star.edu.sg (P.J.H.); 2Saw Swee Hock School of Public Health, National University of Singapore, Singapore 117549, Singapore; ephbamh@nus.edu.sg (M.H.); ephnkmr@nus.edu.sg (N.K.B.M.R.); 3Department of Surgery, Yong Loo Lin School of Medicine, National University of Singapore, Singapore 117597, Singapore; 4Department of Surgery, National University Hospital, Singapore 119054, Singapore; 5Department of General Surgery, Tan Tock Seng Hospital, Singapore 308433, Singapore; ern_yu_tan@ttsh.com.sg; 6Lee Kong Chian School of Medicine, Nanyang Technology University, Singapore 308232, Singapore; 7Division of Medical Oncology, National Cancer Centre Singapore, Singapore 169610, Singapore; elaine.lim.hsuen@singhealth.com.sg; 8Department of Cancer Informatics, National Cancer Centre Singapore, Singapore 169610, Singapore; phyu.n@nccs.com.sg; 9Division of Radiation Oncology, National Cancer Centre Singapore, Singapore 169610, Singapore; wong.fuh.yong@singhealth.com.sg (F.Y.W.)

**Keywords:** breast cancer, chemotherapy, febrile neutropenia, epidemiology, ethnicity

## Abstract

**Simple Summary:**

In this two-part study, we assessed the risk factors associated with FN. On top of commonly associated risk factors such as age and body mass index, we found that there are ethnic differences in terms of FN incidence. Notably, patients of non-Chinese ethnicity had a higher FN incidence. This trend remained significant even after multiple adjustments with other patients, tumors, and treatment characteristics. Further analysis exploring differences across ethnicities revealed that Indian patients were more likely to experience greater dips in absolute neutrophil count during chemotherapy and Malay patients were more likely to be administered with anthracyclines, both of which contribute to higher FN risk. Importantly, we also found that non-Chinese patients were more likely to experience multiple FN episodes during chemotherapy. Additional research is needed to investigate the possibility of pharmacogenetic differences across these ethnicities.

**Abstract:**

The majority of published findings on chemotherapy-induced febrile neutropenia (FN) are restricted to three ethnic groups: Asians, Caucasians, and African Americans. In this two-part study, we examined FN incidence and risk factors in Chinese, Malay, and Indian chemotherapy-treated breast cancer (BC) patients. Hospital records or ICD codes were used to identify patients with FN. In both the Singapore Breast Cancer Cohort (SGBCC) and the Joint Breast Cancer Registry (JBCR), the time of the first FN from the start of chemotherapy was estimated using Cox regression. Multinomial regression was used to evaluate differences in various characteristics across ethnicities. FN was observed in 170 of 1014 patients in SGBCC. The Cox model showed that non-Chinese were at higher risk of developing FN (HR_Malay_ [95% CI]:2.04 [1.44–2.88], *p* < 0.001; HR_Indian_:1.88 [1.11–3.18], *p* = 0.018). In JBCR, FN was observed in 965 of 7449 patients. Univariable Cox models identified ethnicity, a lower baseline absolute neutrophil count, non-luminal A proxy subtypes, and anthracycline-containing regimens as risk factors. Disparities across ethnicities’ risk (HR_Malay_:1.29 [1.07–1.54], *p* = 0.006; HR_Indian_:1.50 [1.19–1.88], *p* < 0.001) remained significant even after further adjustments. Finally, an age-adjusted multinomial model showed that Malays (*p =* 0.006) and Indians (*p* = 0.009) were significantly more likely to develop multiple episodes of FN during treatment. Ethnic differences in chemotherapy-induced FN among BC patients exist. Further studies can focus on investigating pharmacogenetic differences across ethnicities.

## 1. Introduction

Febrile neutropenia (FN) is a prominent hematological adverse effect of chemotherapy that leads to dosage decrease and treatment delay [1]. Both are associated with a decrease in relative dose intensity, which hinders the recovery of early-stage breast cancer patients [1,2,3]. Furthermore, FN is a potentially fatal toxicity, akin to sepsis, and may necessitate emergency admission and hospitalization [4,5].

The occurrence of chemotherapy-induced FN may be associated with ethnic origin, with Asians being more vulnerable to FN than patients of European ancestry [6,7,8,9,10,11,12]. A review done by O’Donnell found that East Asians are more susceptible to chemotherapy side effects, such as neutropenia [13]. In conjunction, a meta-analysis exploring the incidence of docetaxel-induced severe neutropenia between Asian and non-Asian clinical studies found notable differences between the two populations [6]. Another multi-center study reported that racial differences in acute toxicity exist in breast cancer patients treated with fluorouracil, epirubicin, and cyclophosphamide (FEC) 100 chemotherapy [8]. It should be highlighted that the majority of published findings on cancer pharmaco-ethnicity are restricted to three ethnic groups: Asians, Caucasians, and African Americans [13]. Hence, there is a need to further explore the prevalence and risk factors of FN in other ethnic groups or geographic settings [7,11,14].

In this two-part study, we first examined the incidence of chemotherapy-induced FN in a retrospective cohort study comprising 1014 breast cancer patients in Singapore. Using the independent Joint Breast Cancer Registry, we further studied the incidence of FN amongst 7449 chemotherapy-treated patients as well as associations between ethnicity and other patients, tumor, and treatment characteristics.

## 2. Materials and Methods

This study comprises two independent datasets from three health clusters in Singapore: National University Health System (NUHS), National Healthcare Group (NHG), and Singapore Health Services (SingHealth). We explored the associations between ethnicity and FN occurrence in breast cancer patients recruited from NUHS and NHG, using breast cancer patients recruited from the Singapore Breast Cancer Cohort (SGBCC) (*n* = 1014). To reach a deeper understanding of our findings, we studied various potential confounders in a larger registry dataset from the SingHealth cluster, the Joint Breast Cancer Registry (JBCR) (*n* = 7449).

### 2.1. Singapore Breast Cancer Cohort (SGBCC) Dataset

Briefly, SGBCC is a multi-institutional breast cancer cohort established in 2009. The seven recruiting public hospitals, namely, National University Hospital (NUH), KK Women’s and Children’s Hospital (KKH), Tan Tock Seng Hospital (TTSH), National Cancer Centre Singapore (NCCS), Singapore General Hospital (SGH), Changi General Hospital (CGH), and Ng Teng Fong General Hospital (NTFGH), collectively treat ~76% of all breast cancer patients in Singapore; the details on the cohort profile can be found in this paper [15]. Patients from NUH and TTSH were included in the analytical cohort. FN cases were identified via hospital admission date where the laboratory test has a neutrophil count of fewer than 500 cells/μL together with a single oral temperature greater than or equal to 38.3 °C or a temperature greater than or equal to 38 °C for at least an hour during or within 30 days from the last chemotherapy session. Univariable Cox regression models were used to assess the associations between the various risk factors and FN. Multivariable Cox regression included statistically significant variables from the univariable model as well as conventional risk factors, such as age and disease stage.

SGBCC was approved by the National Healthcare Group Domain Specific Review Board (reference number: 2009/00501) and the SingHealth Centralised Institutional Review Board (CIRB Ref: 2019/2246 [2010/632/B]). Informed consent was obtained from all patients.

### 2.2. Joint Breast Cancer Registry (JBCR)

The Joint Breast Cancer Registry (JBCR) is a hospital-based cancer registry that holds a wealth of information on breast cancer patients diagnosed or treated at hospitals within the SingHealth Cluster in Singapore [16]. The sites include SGH, NCCS, KKH, CGH, and Seng Kang General Hospital (SKH). Data collected by JBCR include patient demographics, primary tumor information (e.g., diagnosis date, tumor histology, stage at diagnosis, tumor size, nodal involvement, hormonal status), treatment details, adverse drug reactions observed, survival status, and treatment-related costs.

JBCR was approved by the SingHealth Centralized Institutional Review Board (CIRB Ref: CIRB2019/2419 and 2016/3010). As the study relies on deidentified registry-based data, the need to obtain informed consent was waived. This study was approved by the A*STAR Institutional Review Board (2020–005).

### 2.3. Exclusions

Between 2010 and 2016, a total of 10,415 patients were recruited to be part of SGBCC. Patients from overlapping institutions (*n* = 5953) and patients without information on febrile neutropenia (*n* = 2953) were excluded. Between 2005 and 2020, a total of 10,120 patients were registered in the JBCR. For both datasets, we excluded male patients (SGBCC = 0, JBCR = 24), patients without known race (SGBCC = 0, JBCR = 5), patients of “Other” race (SGBCC = 43, JBCR = 1150), patients with previous chemotherapy (chemotherapy that occurred before the recorded diagnosis date) (SGBCC = 2, JBCR = 37), and patients without chemotherapy (SGBCC = 44, JBCR = 262). Additionally, patients who underwent both neoadjuvant and adjuvant chemotherapy but had their first FN occur in the latter were excluded (SGBCC = 0, JBCR = 14). Finally, patients with missing stages or diagnosed with DCIS or stage IV breast cancer were also excluded (SGBCC = 406, JBCR = 1179). The analytical cohort comprised 1014 SGBCC and 7449 JBCR breast cancer patients (Appendix A and Figure 1).

### 2.4. Outcomes of Interest

In JBCR, patients with FN were identified either via hospital admission date or Accident and Emergency visit date with confirmation of ICD FN diagnosis (fever: ICD9/10:2880/D70 and neutropenia: ICD9/10:7806/R509). The primary outcome of interest was the incidence of FN during or within 30 days from the last chemotherapy session. The secondary outcome of interest was the total number of FN occurrences during the same time period. Details on the definition of study time points can be found in Appendix A.

### 2.5. Patient Characteristics

Patient characteristics that were included in this analysis were treatment institution, age at diagnosis, year of diagnosis, ethnicity (Chinese/Malay/Indian), baseline and changes in body mass index (BMI), and absolute neutrophil count (ANC). Information on time points of BMI and ANC records used is given in Appendix A. It should be noted that a large proportion of participants had incomplete records for baseline BMI, changes in BMI, baseline ANC, and changes in ANC.

### 2.6. Tumor Characteristics

Tumor characteristics considered include stage at diagnosis (stage I/II/III), estrogen receptor (ER) status (yes/no), progesterone receptor (PR) status (yes/no), human epidermal growth factor receptor 2 (HER2) status (yes/no), tumor size (≤2 cm/>2–5 cm/>5 cm/missing), and histological grade (well-/moderately/poorly differentiated). Intrinsic-like subtypes were defined using immunohistochemical markers for ER, PR, and HER2 in conjunction with histologic grade: luminal A [ER+/PR+, HER2−, well- or moderately differentiated]/luminal B [HER2−] (ER+/PR+, HER2−, and poorly differentiated)/luminal B [HER2+] (ER+/PR+, HER2+, and poorly differentiated)/HER2-overexpressed [HER2+]/triple-negative [ER−, PR−, and HER2−] [17].

### 2.7. Treatment Details

Treatment characteristics included surgery type (mastectomy/breast conserving/unknown/no surgery/missing), radiotherapy (yes/no/missing), tamoxifen treatment (yes/no/missing), herceptin treatment (yes/no/missing), chemotherapy type (neoadjuvant/adjuvant), and granulocyte colony-stimulating factor (GCSF) use (yes/no). Individual GCSF time points (before/during chemotherapy) were derived based on the recorded administration date. The variable “GCSF before FN” indicates the administration of GCSF before chemotherapy and/or GCSF given during the chemotherapy period but before any occurrence of FN. Drug types used in chemotherapy were also included, and more information can be found in Appendix A.

### 2.8. Statistical Analyses

Descriptive analyses were conducted to characterize patient, tumor, and treatment characteristics. The associations between FN and the various characteristics were studied using the Chi-square test and Kruskal–Wallis test for categorical and continuous variables, respectively.

Univariable and multivariable Cox regression models were used to assess the association between the various risk factors and FN (survival package in R, where the *Surv (time to event, FN event)* command was used to estimate hazard ratios (HR) and corresponding 95% confidence intervals (CI)). For patients with FN, "time to event" was defined as the time from the start of chemotherapy to the first FN occurrence. For patients without FN, “time to event” was defined as the time from the start of chemotherapy to the end of chemotherapy. In view of the length of chemotherapy treatment courses, time to event was truncated at 210 days (6 months of chemotherapy + 30 days). Patients with missing values for continuous variables were not included in the univariate analysis. In the multivariable Cox regression model, the model was adjusted for all factors significantly associated with FN risk in the univariable model. *Cox.zph* was used to test for the proportionality of the Cox regression model fits. In addition, the Kaplan–Meier (KM) method was used to analyze FN risk by stage and GCSF status subsets (survival package in R); survival curves were compared using the log-rank test.

To assess patient, tumor, and treatment differences across ethnicities, multinomial logistic regression models were used, adjusting for age at diagnosis. Linear regression models for patients with complete records were also used to evaluate differences in ANCs across ethnicities, adjusted for age at diagnosis and BMI. A test of equal or given proportions for missingness in baseline BMI was conducted using *prop.test*. A Sensitivity analysis, including patients with missing baseline BMIs, was done.

Finally, to evaluate how different patient, tumor, and treatment characteristics might interact with ethnicity to modify the risk of FN, two-way interaction terms were created and assessed.

## 3. Results

### 3.1. FN-Associated Risk Factors in SGBCC

Among 1014 breast cancer patients from NUH and TTSH, FN occurred in 170 patients (16.7%). The mean age at diagnosis was 54 years (IQR: 48.0 to 59.7). The ethnic makeup of this chemotherapy-treated breast cancer population was 77.0%, 16.5%, and 6.5% for Chinese, Malay, and Indian, respectively.

Univariate Cox model showed that compared to Chinese patients, Malay patients were at higher risk of developing FN (HR_Malay_ [95% CI]: 2.04 [1.44–2.88], *p* < 0.001; HR_Indian_: 1.88 [1.11–3.18], *p* = 0.018) (Appendix A). Trends in associations between ethnicity and FN risk remained unchanged even after accounting for age at diagnosis, cancer stage, surgery type, and adjuvant trastuzumab treatment (HR_Malay_: 2.03 [1.42–2.91], *p* < 0.001; HR_Indian_: 1.99 [1.17–3.40], *p* = 0.012) (Appendix A).

### 3.2. Patient Characteristics in JBCR

Table 1 shows the descriptive statistics of patients’ characteristics. Among the 7449 patients included in the JBCR, 6484 (87.0%) did not experience FN, and the remaining 965 (13.0%) were diagnosed with FN at least once during their chemotherapy treatment. The mean age at diagnosis was 54 years (IQR: 47.0 to 61.0), and the majority of the patients were of Chinese ethnicity (80.5%). The mean baseline BMI was 23.5 (IQR: 20.6 to 27.0) and the mean baseline ANC was 4.0 10^9^/L (IQR: 3.1 10^9^/L to 5.1 10^9^/L). The most frequently used chemotherapy regimen was taxanes in combination with anthracyclines and cyclophosphamides (54.2%). A total of 1545 (20.7%) patients were given GCSF before FN. Of the 965 first occurrences of FN, 497 (51.5%) occurred within the first month of chemotherapy (Figure 2).

### 3.3. Ethnicity Is Associated with FN Occurrence after Adjusting for Patient, Tumor, and Treatment Characteristics

In univariate Cox models, those of non-Chinese ethnicity, diagnosed before 2008, older at diagnosis, greater maximum decrease in BMI, lower baseline ANC, negative ER status, triple-negative subtype, and anthracycline-containing regimens were significantly associated with FN occurrence (Table 2, refer to Appendix A for additional adjusted models). Additionally, patients who underwent additional treatments such as radiotherapy, herceptin, or tamoxifen treatments, had a significantly lower risk of developing FN. Further adjustments for patient, tumor and treatment variables that were significant in the univariate cox models did not result in any appreciable change in the hazard ratios and ethnicity remained as a significant factor contributing to FN risk (HR_Malay_:1.29 [1.07–1.54], *p* = 0.006; HR_Indian_:1.50 [1.19–1.88], *p* < 0.001).

Figure 3a presents the overall Kaplan–Meier curve for FN occurrence among the 7449 BC patients. In subset analysis, according to the stage at diagnosis, ethnic differences in risk for FN persisted, where Malays and Indians were at higher risk for FN as compared to Chinese, even though the effect sizes were attenuated and differences were only significant for stage II (HR_Malay_: 1.30 [0.99–1.70], *p* = 0.058; HR_Indian_: 1.51 [1.10–2.09], *p* = 0.012; log-rank *p*: 0.011) patients (Figure 3b–d).

Further subset analysis was done according to the presence or absence of GCSF before FN. Among those who received GCSF before FN, Malay patients displayed a higher risk for FN (HR_Malay_: 1.39 [1.02–1.88], *p* = 0.034; log-rank *p*: 0.053) (Figure 4a). On the other hand, among those who did not receive GCSF before FN, Indian patients showed a higher risk for FN (HR_Indian_: 1.56 [1.18–2.07], *p* = 0.002; log-rank *p*: 0.015) (Figure 4b).

### 3.4. Tumor and Treatment Characteristics Differ across Ethnicities

Table 3 shows age-adjusted associations between ethnicity and various patient, tumor, and treatment characteristics in multinomial regression models. The result of the multinomial regression model can be interpreted as follows: the odds of a Malay patient developing stage III breast cancer is 2.3 times that of a Chinese patient. Apart from treatment institution and diagnosis year, patient characteristics were not significantly different between ethnic groups. Compared to the referent Chinese group, Malays, and Indians were more likely to be diagnosed with late-stage cancers and larger tumor sizes. When looking at proxy subtypes, Malays were at higher odds of developing luminal B (HER2-neg) and HER2-overexpressed cancers, while Indians were at higher odds of developing triple-negative cancers. Additionally, among those not prescribed GCSF before chemotherapy, even though Indian patients have a higher baseline ANC than Chinese (Beta_Indian:_ 0.51 [0.29–0.73], *p* < 0.001), they also experienced a greater drop in ANC during chemotherapy (Beta_Indian:_ 0.41 [0.16–0.67], *p* = 0.001) (Table 4). Similar observations were seen in Malay patients, though the results were not significant. Further investigations using the proportions test showed that the missingness for BMI across ethnicities was not at random (*p* < 0.001). A sensitivity analysis including patients with missing BMI as a separate category was done. The association between ethnicity and baseline ANC became more significant (Beta_Malay_ [95% CI]: 0.24 [0.09–0.38], *p* = 0.001; Beta_Indian_: 0.65 [0.46–0.84], *p* < 0.001) (Appendix A). This more pronounced association was also seen in the magnitude of the ANC decrease (Beta_Malay_: 0.23 [0.07–0.40], *p* = 0.004; Beta_Indian_: 0.50 [0.29–0.71], *p* < 0.001) (Appendix A).

In terms of treatment, Malays and Indians were less likely to have undergone tamoxifen and radiotherapy. Additionally, Malays were more likely to have been administered anthracyclines but less likely to be prescribed GCSF before FN (Table 3).

### 3.5. Various Tumor and Treatment Characteristics Did Not Differentially Affect FN Risk across Ethnicities

Additional models were constructed to further evaluate the interaction between ethnicity and other significantly associated variables in the univariable Cox models (Appendix A). No key interactions were observed.

### 3.6. Non-Chinese Breast Cancer Patients More Frequently Developed Multiple FN Episodes

We further investigated the total number of FN episodes that occurred during the chemotherapy treatment period. The age-adjusted multinomial model showed that both Malay (OR_Malay_ [95%CI]: 3.65 [1.45–9.16], *p* = 0.006) and Indian patients (OR_Indian_: 4.43 [1.45–13.60], *p* = 0.009) were significantly more likely to develop more than 3 episodes of FN during treatment as compared to Chinese patients (Table 3).

## 4. Discussion

In the SGBCC population, comprising 1014 chemotherapy-treated breast cancer patients, the incidence rate of FN was 16.7%. Notably, patients of non-Chinese ethnicity had a higher FN incidence. The ethnic disparity was also seen in an independent JBCR hospital-register-based population of 7449 chemotherapy-treated breast cancer patients, where 13% of patients experienced FN episodes. In JBCR, non-Chinese were estimated to be at a 1.5 times elevated risk of developing chemotherapy-induced FN. Patients of non-Chinese ethnicity also more frequently experienced multiple FN episodes. Other significant factors associated with FN include age at diagnosis, anthracycline-containing regimens, and the absence of non-chemotherapy treatments, such as radiotherapy, herceptin, and tamoxifen treatment. While exploring differences across ethnicities, we found that Malays and Indians were more likely to be diagnosed at a later stage. These two groups were also less likely to undergo additional treatments, more likely to be prescribed with an anthracycline-containing regimen, and experienced greater dips in ANC during chemotherapy, which may contribute to the elevated FN risk. However, the ethnic disparity remained even after significantly associated patient, tumor, and treatment characteristics were adjusted for.

Previously reported patient-specific FN risk factors among Asian breast cancer patients include low BMI (BMI < 23 kg/m^2^), low baseline white blood cell count, and low ANC [18,19,20]. In addition, age, poor performance status, and particular genetic variants have all been linked to elevated FN risk [20]. Similarly, our study showed that age and low baseline ANC were risk factors for FN.

Our study observed that anthracycline-containing regimens placed patients at higher risk for FN. However, this trend differs from what is commonly observed, where studies have shown that a taxane-based regimen contributes to higher FN incidence [21,22,23]. It should be noted that most of these studies focused on Caucasian populations, which may explain the differences in toxicities observed when compared to our Asian population. A Nigerian study comprising 113 breast cancer patients reported the lack of a significant relationship between taxane-based regimens and FN risk [24]. The authors attributed this finding to the fact that the majority of the patients were first started on anthracycline-based treatments, but 36% were later switched to a taxane-based regimen [24]. By chemotherapy regimen in Chen et al.’s study in Singapore, FN incidence observed in those treated with anthracycline- and taxane-based drugs was 15.0% and 11.7%, respectively [25], which concurs with our finding. Additionally, our study also found that compared to patients who received a combination of only anthracyclines and cyclophosphamides, those who received taxanes, anthracyclines, and cyclophosphamides were at lower risk for FN, even though this association was not significant. In agreement, a meta-analysis of 15 randomized trials by Zheng et al. reported that combined cytotoxic regimens led to significantly fewer FN events than single agents [26].

Ethnicity has been suggested to be associated with FN incidence previously [6,7,8,9,11,12]. Ethnic variance in treatment toxicity is widely acknowledged as a key element explaining inter-individual variation in cancer treatment responsiveness [13]. East Asians have been reported to be more susceptible to toxic side effects of chemotherapy than their Caucasian counterparts, requiring dose reduction in many studies [13]. In an earlier work by Chen et al. on 583 Asian cancer patients in Singapore, it was reported that FN incidence in Chinese, Malay, and Indian patients was 15.6%, 24.5%, and 25%, respectively [25]. In agreement, the results from our study showed that both Malay and Indian patients were at higher risk of FN, where FN incidence in Chinese, Malay, and Indian patients was 12.3%, 14.6%, and 18.1%, respectively.

In local clinical practice, patients diagnosed with locally advanced disease are prescribed anthracycline in combination with taxanes, while those with early-stage disease are given taxanes only. Similarly, the Malay patients in our group were observed to be diagnosed at a higher stage and were more likely to be given anthracycline in addition to taxanes, which can explain their greater FN risk. However, even after accounting for significantly associated patient, tumor, and treatment factors, the disparity between the effect of chemotherapy on FN risk persisted across ethnicities, indicating the possibility that other unidentified contributors exist. In our study, we found that Indian patients experienced larger dips in ANC during chemotherapy, which prompts further investigation in this area. Furthermore, inter-individual and ethnic differences in how docetaxel and doxorubicin are metabolized, as well as interethnic disparities in docetaxel- and doxorubicin-induced myelosuppression, are shown to impact the results of treatment [10,27,28]. Additionally, previous studies have shown that patients with comorbidities, such as renal and liver diseases, were at higher risk of systemic toxicities, such as FN [25,29]. This, combined with the widely observed trend that Malays and Indians are more likely to suffer from chronic conditions [30,31,32], can explain the higher FN incidence experienced. However, data on comorbidities were not properly collected for this study, which limited further investigation in this area.

This study found a consistent ethnic disparity in chemotherapy-induced FN in two large datasets comprising Chinese, Malay, and Indian breast cancer patients. However, several limitations should be noted. Due to the retrospective design of this study, selection bias cannot be eliminated. Another limitation is the incompleteness of clinical records, such as detailed information on comorbidities and chemotherapy regimens. The dataset has a sizable proportion of subjects (28%) with missing baseline BMI values. However, excluding patients with missing baseline BMI biased the results for absolute neutrophil count towards the null. Furthermore, commonly studied outcomes such as FN incidence during each cycle or FN risk due to differences in regimen dosage were unable to be studied due to the lack of information on the chemotherapy cycle or complete regimen details. Although efforts were made to adjust for several different treatment and demographic factors associated with FN, likely, there is still significant residual confounding in the results.

## 5. Conclusions

In summary, the results of this study suggest that there are ethnic differences in the risk of chemotherapy-induced FN. The findings present an opportunity to improve the precision of breast cancer treatment. Taking into consideration the susceptibility of certain ethnic groups to adverse effects, prophylactic or alternative treatments can be utilized to avoid these side effects. Moreover, close monitoring for early signs of infection may be emphasized in high-risk ethnic populations. Additional research where data for treatment regimens are properly collected is needed to investigate the possibility of pharmacogenetic differences across these ethnicities and formulate better-tailored treatment combinations. Further studies can also explore differences in the effects of FN across ethnicities, such as the need for reductions in relative dose intensity or delays in chemotherapy schedules, as well as survival outcomes.

## Figures and Tables

**Figure 1 cancers-15-03590-f001:**
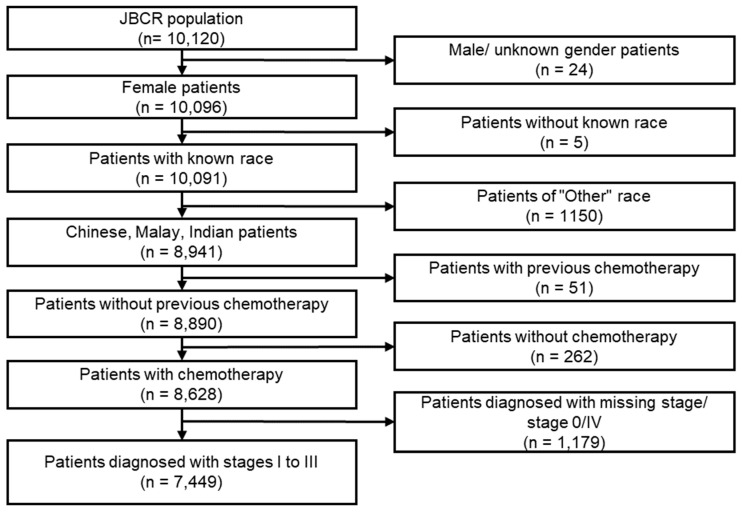
Flow chart of JBCR study population.

**Figure 2 cancers-15-03590-f002:**
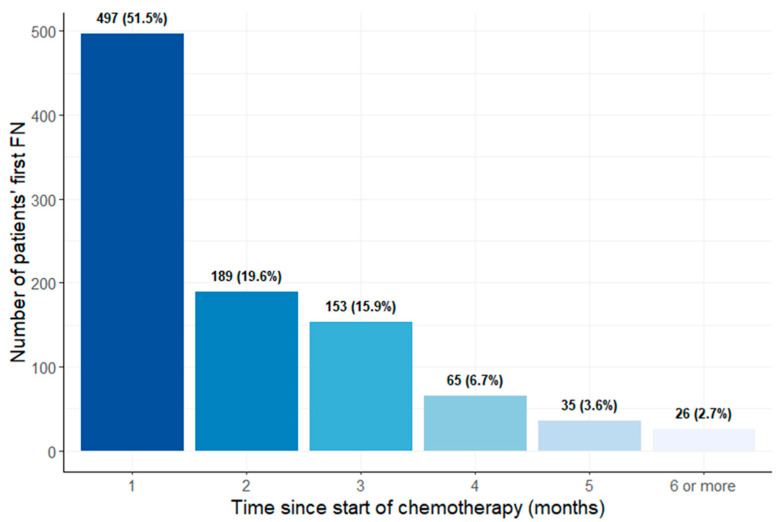
Bar plot showing the spread of first FN occurrence throughout chemotherapy treatment.

**Figure 3 cancers-15-03590-f003:**
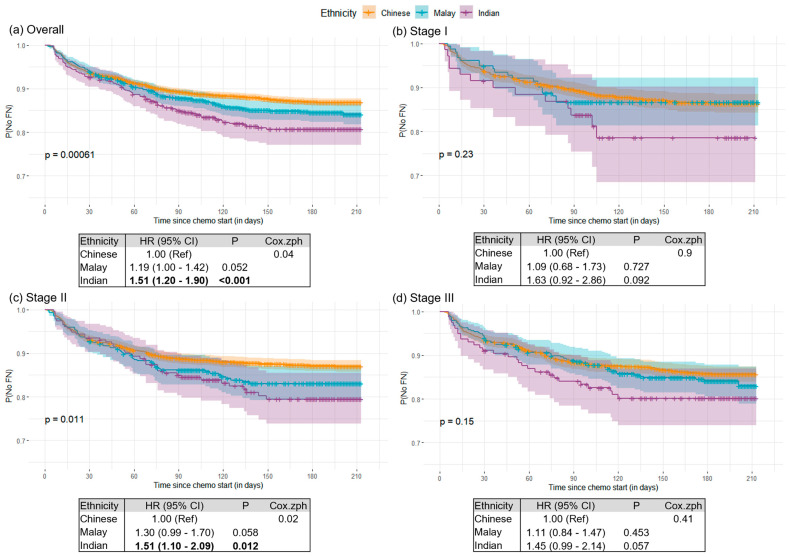
Kaplan–Meier plots for (**a**) overall population (*n* = 7449), (**b**) stage I (*n* = 1570), (**c**) stage II (*n* = 3570), and (**d**) stage III (*n* = 2309). Time to first FN is illustrated according to ethnicity (Chinese, Malay, Indian). The in-model *p*-value is a log-rank test. Tables below each figure show associations of ethnicity with FN occurrence. Hazard ratios (HR) and 95% confidence intervals (CI) were estimated using the Cox regression models. Bold indicates statistical significance at *p*-value < 0.05.

**Figure 4 cancers-15-03590-f004:**
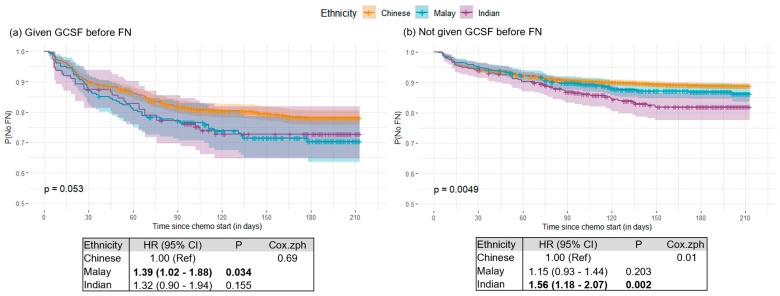
Kaplan–Meier plots for (**a**) patients who were given granulocyte colony-stimulating factor (GCSF) before any FN (*n* = 1545) and (**b**) patients who were not given GCSF before any FN (*n* = 5904). Tables below each figure show associations of ethnicity with FN occurrence. Hazard ratios (HR) and 95% confidence intervals (CI) were estimated using the Cox regression models. Bold indicates statistical significance at *p*-value < 0.05.

**Table 1 cancers-15-03590-t001:** Characteristics of the study population (*n* = 7449). *p*-value (*p*) for categorical variables is based on the Chi-square test and *p* for continuous variables are based on the Kruskal–Wallis test.

	Total, *n* = 7449	0, *n* = 6484	1, *n* = 965	*p*-Value
Patient characteristics				
Ethnicity				
Chinese	5993 (80.5)	5257 (81.1)	736 (76.3)	<0.001
Malay	998 (13.4)	852 (13.1)	146 (15.1)	
Indian	458 (6.1)	375 (5.8)	83 (8.6)	
Institution ^a^				
NCCS	3079 (41.3)	2685 (41.4)	394 (40.8)	<0.001
CGH	891 (12.0)	837 (12.9)	54 (5.6)	
KKH	1478 (19.8)	1262 (19.5)	216 (22.4)	
SGH	1936 (26.0)	1646 (25.4)	290 (30.1)	
SKH	65 (0.9)	54 (0.8)	11 (1.1)	
Age at diagnosis (years, IQR ^b^)	54.0 (47.0–61.0)	53.0 (46.0–60.0)	54.0 (47.0–61.0)	0.005
Menopausal status				
Premenopausal	2554 (34.3)	2239 (34.5)	315 (32.6)	0.003
Postmenopausal	3142 (42.2)	2667 (41.1)	475 (49.2)	
Missing	1753 (23.5)	1578 (24.3)	175 (18.1)	
Baseline Body-Mass Index (BMI) (kg/m^2^, IQR)	23.6 (21.7–27.1)	23.6 (22.1–26.8)	23.6 (21.7–27.1)	0.598
Unknown (*n*)	2088	1847	241	
Maximum BMI decrease (kg/m^2^, IQR)	1.0 (0.2–2.0)	0.7 (0.1–1.5)	1.0 (0.2–2.0)	<0.001
Unknown (*n*)	4412	3867	545	
Baseline absolute neutrophil count (10^9^/L, IQR)	4.2 (3.5–4.7)	4.2 (3.7–4.6)	4.2 (3.5–4.7)	0.042
Unknown (*n*)	2661	2360	301	
Tumor characteristics				
Stage				
I	1570 (21.1)	1380 (21.3)	190 (19.7)	0.197
II	3570 (47.9)	3117 (48.1)	453 (46.9)	
III	2309 (31.0)	1987 (30.6)	322 (33.4)	
Estrogen receptor status				
No	2292 (30.8)	1964 (30.3)	328 (34.0)	0.028
Yes	5020 (67.4)	4397 (67.8)	623 (64.6)	
Missing	137 (1.8)	123 (1.9)	14 (1.5)	
Progesterone receptor status				
No	3056 (41.0)	2630 (40.6)	426 (44.1)	0.049
Yes	4254 (57.1)	3729 (57.5)	525 (54.4)	
Missing	139 (1.9)	125 (1.9)	14 (1.5)	
HER2 status				
No	4684 (62.9)	4067 (62.7)	617 (63.9)	0.618
Yes	2522 (33.9)	2201 (33.9)	321 (33.3)	
Missing	243 (3.3)	216 (3.3)	27 (2.8)	
Grade				
Well-differentiated	409 (5.5)	360 (5.6)	49 (5.1)	0.009
Moderately differentiated	2575 (34.6)	2279 (35.1)	296 (30.7)	
Poorly differentiated	4152 (55.7)	3570 (55.1)	582 (60.3)	
Missing	313 (4.2)	275 (4.2)	38 (3.9)	
Tumor size (cm)				
≤2	2470 (33.2)	2119 (32.7)	351 (36.4)	0.534
>2–5	3319 (44.6)	2876 (44.4)	443 (45.9)	
>5	615 (8.3)	536 (8.3)	79 (8.2)	
Missing	1045 (14.0)	953 (14.7)	92 (9.5)	
Subtype				
Luminal A	2153 (28.9)	1906 (29.4)	247 (25.6)	0.12
Luminal B (HER2-neg)	1676 (22.5)	1458 (22.5)	218 (22.6)	
Luminal B (HER2-pos)	118 (1.6)	101 (1.6)	17 (1.8)	
HER2-overexpressed	958 (12.9)	831 (12.8)	127 (13.2)	
Triple-negative	1018 (13.7)	868 (13.4)	150 (15.5)	
Missing	1526 (20.5)	1320 (20.4)	206 (21.3)	
Treatment characteristics				
Surgery type				
Mastectomy	5075 (68.1)	4387 (67.7)	688 (71.3)	0.037
Breast conserving	1959 (26.3)	1727 (26.6)	232 (24.0)	
No surgery	56 (0.8)	53 (0.8)	3 (0.3)	
Missing	359 (4.8)	317 (4.9)	42 (4.4)	
Chemotherapy type				
Neoadjuvant	1277 (17.1)	1062 (16.4)	215 (22.3)	<0.001
Adjuvant	6172 (82.9)	5422 (83.6)	750 (77.7)	
Chemotherapy combination ^c^				
TAC	4040 (54.2)	3423 (52.8)	617 (63.9)	<0.001
TC	1643 (22.1)	1489 (23.0)	154 (16.0)	
AC	872 (11.7)	743 (11.5)	129 (13.4)	
T++	615 (8.3)	567 (8.7)	48 (5.0)	
A++/C++/TA/others/missing	279 (3.7)	262 (4.0)	17 (1.8)	
Granulocyte colony-stimulating factor (GCSF) overall				
No	5504 (73.9)	5270 (81.3)	234 (24.2)	<0.001
Yes	1945 (26.1)	1214 (18.7)	731 (75.8)	
GCSF before FN				
No	5904 (79.3)	5272 (81.3)	632 (65.6)	<0.001
Yes	1545 (20.7)	1212 (18.7)	333 (34.5)	

^a^ NUH = National University Hospital, KKH = KK Women’s and Children’s Hospital, TTSH = Tan Tock Seng Hospital, NCCS = National Cancer Centre Singapore, SGH = Singapore General Hospital, CGH = Changi General Hospital, NTFGH = Ng Teng Fong General Hospital, SKH = SengKang General Hospital. ^b^ IQR = Inter-quartile range. ^c^ TAC = taxanes+ anthracyclines+ cyclophosphamides, TC = taxanes+ cyclophosphamides, AC = anthracyclines+ cyclophosphamides, TA = taxanes+ anthracyclines, T++ = taxanes and other drugs (not including anthracyclines/cyclophosphamides), A++ = anthracyclines and other drugs (not including taxanes and cyclophosphamides), C++ = cyclophosphamides and other drugs (not including taxanes and anthracyclines).

**Table 2 cancers-15-03590-t002:** Associations of variables with FN occurrence (*n* = 7449). Hazard ratios (HR) and 95% confidence intervals (CI) were estimated using the Cox regression models. Values are considered statistically significant when *p*-value < 0.05.

	Univariate			Adjusted for Significant Patient, Tumor, and Treatment Characteristics		
	HR (95% CI)	*p*	Cox.zph	HR (95% CI)	*p*	Cox.zph
Patient characteristics						
Ethnicity						
Chinese	1.00 (Ref)		0.04	1.00 (Ref)		0.04
Malay	1.19 (1.00–1.42)	0.052		1.29 (1.07–1.54)	0.006	
Indian	1.51 (1.20–1.90)	<0.001		1.50 (1.19–1.88)	<0.001	
Institution ^a^						
NCCS	1.00 (Ref)		0.19	1.00 (Ref)		0.09
CGH	0.44 (0.33–0.58)	<0.001		0.44 (0.33–0.58)	<0.001	
KKH	1.15 (0.98–1.36)	0.091		0.89 (0.75–1.07)	0.217	
SGH	1.16 (1.00–1.35)	0.05		1.18 (1.01–1.37)	0.043	
SKH	1.28 (0.71–2.34)	0.414		0.97 (0.53–1.79)	0.93	
Year of diagnosis						
Before 2008	1.00 (Ref)		0	1.00 (Ref)		<0.001
During or after 2008	0.84 (0.72–0.99)	0.034		0.83 (0.69–0.99)	0.043	
Age at diagnosis	1.01 (1.00–1.01)	0.044	0.58	1.01 (1.01–1.02)	<0.001	0.53
Baseline Body-Mass Index (BMI, kg/m^2)^	1.00 (0.98–1.01)	0.837	0.01			
Maximum BMI decrease (kg/m^2^)	1.14 (1.09–1.20)	<0.001	0.31			
Baseline absolute neutrophil count (10^9^/L)	0.94 (0.89–0.99)	0.018	0.03			
Tumor characteristics						
Stage						
I	1.00 (Ref)		0.19	1.00 (Ref)		0.2
II	1.02 (0.86–1.20)	0.86		0.86 (0.72–1.03)	0.102	
III	1.08 (0.90–1.29)	0.427		0.81 (0.67–0.99)	0.038	
Estrogen receptor status						
No	1.00 (Ref)		0.74			
Yes	0.87 (0.76–0.99)	0.036				
Missing	0.75 (0.44–1.27)	0.283				
Subtype						
Luminal A	1.00 (Ref)		0.81	1.00 (Ref)		0.8
Luminal B (HER2-neg)	1.12 (0.94–1.35)	0.21		1.28 (1.06–1.53)	0.009	
Luminal B (HER2-pos)	1.23 (0.75–2.01)	0.412		1.91 (1.16–3.15)	0.011	
HER2-overexpressed	1.12 (0.91–1.39)	0.295		1.97 (1.56–2.49)	<0.001	
Triple-negative	1.30 (1.06–1.59)	0.011		0.97 (0.79–1.19)	0.746	
Missing	1.18 (0.98–1.42)	0.084		1.16 (0.96–1.40)	0.113	
Treatment characteristics						
Surgery type						
Mastectomy	1.00 (Ref)		0.44			
Breast conserving	0.89 (0.77–1.04)	0.141				
No surgery	0.39 (0.13–1.22)	0.107				
Missing	0.90 (0.66–1.23)	0.506				
Radiotherapy						
No	1.00 (Ref)		0.03	1.00 (Ref)		0.038
Yes	0.18 (0.11–0.30)	<0.001		0.25 (0.15–0.43)	<0.001	
Missing	0.32 (0.04–2.24)	0.249		0.29 (0.04–2.10)	0.223	
Tamoxifen treatment						
No	1.00 (Ref)		0.11	1.00 (Ref)		0.097
Yes	0.38 (0.29–0.51)	<0.001		0.50 (0.38–0.67)	<0.001	
Missing	0.37 (0.05–2.60)	0.316		0.41 (0.06–2.93)	0.376	
Herceptin treatment						
No	1.00 (Ref)		0.38	1.00 (Ref)		0.39
Yes	0.32 (0.26–0.40)	<0.001		0.25 (0.20–0.32)	<0.001	
Missing	0.32 (0.12–0.87)	0.025		0.26 (0.10–0.70)	0.007	
Chemotherapy type						
Neoadjuvant	1.00 (Ref)		0.08	1.00 (Ref)		0.19
Adjuvant	0.74 (0.63–0.86)	<0.001		1.02 (0.86–1.21)	0.835	
Chemotherapy combination ^b^						
TAC	1.00 (Ref)		0	1.00 (Ref)		<0.001
TC	0.70 (0.58–0.83)	<0.001		0.62 (0.51–0.76)	<0.001	
AC	1.15 (0.95–1.40)	0.144		0.92 (0.74–1.15)	0.481	
T++	0.53 (0.40–0.72)	<0.001		0.94 (0.69–1.29)	0.712	
A++/C++/TA/others/missing	0.54 (0.33–0.88)	0.013		0.65 (0.40–1.06)	0.087	
Given granulocyte colony-stimulating factor before FN						
No	1.00 (Ref)		0.02	1.00 (Ref)		0.01
Yes	2.07 (1.81–2.36)	<0.001		2.24 (1.93–2.60)	<0.001	

^a^ NUH = National University Hospital, KKH = KK Women’s and Children’s Hospital, TTSH = Tan Tock Seng Hospital, NCCS = National Cancer Centre Singapore, SGH = Singapore General Hospital, CGH = Changi General Hospital, NTFGH = Ng Teng Fong General Hospital, SKH = SengKang General Hospital. ^b^ TAC = taxanes+ anthracyclines+ cyclophosphamides, TC = taxanes+ cyclophosphamides, AC = anthracyclines+ cyclophosphamides, TA = taxanes+ anthracyclines, T++ = taxanes and other drugs (not including anthracyclines/cyclophosphamides), A++ = anthracyclines and other drugs (not including taxanes and cyclophosphamides), C++ = cyclophosphamides and other drugs (not including taxanes and anthracyclines).

**Table 3 cancers-15-03590-t003:** Associations between patient, tumor, and treatment characteristics with ethnicity, adjusted for age at diagnosis. Odds ratios (OR) and 95% confidence intervals (CI) were estimated using multinomial regression. *p* indicates *p*-value obtained from the Wald test. Values are considered statistically significant when *p*-value < 0.05.

	Chinese		Malay			Indian		
	*n*		*n*	OR (95%CI)	*p*	*n*	OR (95%CI)	*p*
Institution ^a^								
NCCS	2489	1.00 (Ref)	395			195		
CGH	652		193	1.93 (1.59–2.34)	<0.001	46	0.92 (0.66–1.28)	0.607
KKH	1151		231	1.20 (1.01–1.44)	0.041	96	1.04 (0.81–1.34)	0.763
SGH	1645		176	0.67 (0.55–0.81)	<0.001	115	0.89 (0.70–1.13)	0.333
SKH	56		3	0.34 (0.11–1.09)	0.07	6	1.37 (0.58–3.22)	0.469
Year of diagnosis								
Before 2008	1073	1.00 (Ref)	153			65		
During or after 2008	4920		845	1.30 (1.08–1.57)	0.005	393	1.37 (1.05–1.80)	0.022
Stage								
I	1343	1.00 (Ref)	156			71		
II	2928		412	1.23 (1.01–1.50)	0.036	230	1.50 (1.14–1.97)	0.004
III	1722		430	2.30 (1.88–2.80)	<0.001	157	1.79 (1.34–2.39)	<0.001
Estrogen receptor status								
No	1832	1.00 (Ref)	296			164		
Yes	4043		688	1.02 (0.88–1.18)	0.781	289	0.78 (0.64–0.96)	0.018
Missing	118		14			5		
Progesterone receptor status								
No	2453	1.00 (Ref)	396			207		
Yes	3420		588	1.00 (0.87–1.15)	0.955	246	0.82 (0.68–1.00)	0.05
Missing	120		14			5		
HER2 status								
No	3753	1.00 (Ref)	631			300		
Yes	2041		334	1.00 (0.87–1.16)	0.962	147	0.92 (0.75–1.12)	0.399
Missing	199		33			11		
Grade								
Well-differentiated	339	1.00 (Ref)	51			19		
Moderately differentiated	2130		293	0.93 (0.68–1.28)	0.662	152	1.28 (0.79–2.10)	0.317
Poorly differentiated	3280		607	1.27 (0.93–1.73)	0.126	265	1.47 (0.91–2.37)	0.117
Missing	244		47			22		
Tumor size (cm)								
≤2	2071	1.00 (Ref)	271			128		
>2–5	2640		461	1.37 (1.16–1.61)	<0.001	218	1.35 (1.08–1.70)	0.009
>5	450		120	2.11 (1.66–2.68)	<0.001	45	1.65 (1.16–2.35)	0.006
Missing	832		146			67		
Subtype								
Luminal A	1773	1.00 (Ref)	259			121		
Luminal B (HER2-neg)	1304		268	1.44 (1.19–1.73)	<0.001	104	1.18 (0.90–1.55)	0.232
Luminal B (HER2-pos)	98		12	0.89 (0.48–1.65)	0.72	8	1.24 (0.59–2.60)	0.577
HER2-overexpressed	772		133	1.28 (1.02–1.61)	0.032	53	1.05 (0.75–1.47)	0.768
Triple-negative	799		125	1.11 (0.88–1.40)	0.368	94	1.76 (1.32–2.33)	<0.001
Missing	1247		201			78		
Surgery type								
Mastectomy	4114	1.00 (Ref)	676			285		
Breast conserving	1548		270	0.98 (0.84–1.14)	0.757	141	1.27 (1.02–1.57)	0.029
Unknown surgery type	180		26	0.87 (0.57–1.32)	0.513	13	1.04 (0.58–1.84)	0.903
No surgery	39		12	2.08 (1.08–4.01)	0.029	5	1.96 (0.76–5.01)	0.162
Missing	112		14			14		
Radiotherapy								
No	5550	1.00 (Ref)	946			429		
Yes	424		51	0.69 (0.51–0.93)	0.016	25	0.76 (0.50–1.14)	0.186
Missing	19		1			4		
Tamoxifen treatment								
No	5208	1.00 (Ref)	909			414		
Yes	768		88	0.65 (0.51–0.82)	<0.001	42	0.68 (0.49–0.95)	0.022
Missing	17		1			2		
Herceptin treatment								
No	4538	1.00 (Ref)	749			355		
Yes	1404		215	0.98 (0.83–1.15)	0.81	98	0.92 (0.73–1.16)	0.473
Missing	51		34			5		
Chemotherapy								
Neoadjuvant	990	1.00 (Ref)	194			93		
Adjuvant	5003		804	0.81 (0.68–0.96)	0.018	365	0.77 (0.61–0.98)	0.034
Taxanes								
No	931	1.00 (Ref)	152			58		
Yes	5062		846	1.05 (0.87–1.27)	0.586	400	1.29 (0.97–1.71)	0.082
Anthracyclines								
No	2123	1.00 (Ref)	248			145		
Yes	3870		750	1.58 (1.35–1.84)	<0.001	313	1.15 (0.94–1.42)	0.173
Cyclophosphamide								
No	724	1.00 (Ref)	82			46		
Yes	5269		916	1.37 (1.08–1.75)	0.01	412	1.16 (0.84–1.59)	0.373
Chemotherapy combination ^b^								
TAC	3146	1.00 (Ref)	626			268		
TC	1382		166	0.61 (0.51–0.74)	<0.001	95	0.81 (0.64–1.04)	0.095
AC	708		119	0.82 (0.67–1.02)	0.074	45	0.74 (0.53–1.02)	0.066
T++	525		53	0.59 (0.43–0.79)	<0.001	37	0.90 (0.63–1.30)	0.582
A++/C++/TA/others/missing	232		34	0.77 (0.53–1.11)	0.162	13	0.67 (0.38–1.19)	0.175
Given granulocyte colony-stimulating factor (GCSF) before FN								
No	4741	1.00 (Ref)	816			347		
Yes	1252		182	0.84 (0.70–0.99)	0.043	111	1.21 (0.96–1.51)	0.1
GCSF overall								
No	4436	1.00 (Ref)	751			317		
Yes	1557		247	0.95 (0.81–1.11)	0.503	141	1.28 (1.04–1.57)	0.021
Total number of FN episodes								
0	5257	1.00 (Ref)	852			375		
1	627		119	1.19 (0.96–1.47)	0.105	62	1.40 (1.06–1.85)	0.02
2	95		20	1.35 (0.83–2.21)	0.227	17	2.57 (1.51–4.35)	<0.001
≥3	14		7	3.65 (1.45–9.16)	0.006	4	4.43 (1.45–13.60)	0.009

^a^ NUH = National University Hospital, KKH = KK Women’s and Children’s Hospital, TTSH = Tan Tock Seng Hospital, NCCS = National Cancer Centre Singapore, SGH = Singapore General Hospital, CGH = Changi General Hospital, NTFGH = Ng Teng Fong General Hospital, SKH = SengKang General Hospital. ^b^ TAC = taxanes+ anthracyclines+ cyclophosphamides, TC = taxanes+ cyclophosphamides, AC = anthracyclines+ cyclophosphamides, TA = taxanes+ anthracyclines, T++ = taxanes and other drugs (not including anthracyclines/cyclophosphamides), A++ = anthracyclines and other drugs (not including taxanes and cyclophosphamides), C++ = cyclophosphamides and other drugs (not including taxanes and anthracyclines).

**Table 4 cancers-15-03590-t004:** Complete records analysis of associations between absolute neutrophil count (ANC) and ethnicity among patients without GCSF before the start of chemotherapy, adjusted for age at diagnosis, and baseline BMI. Beta values and 95% confidence intervals (CI) were estimated using the linear regression models. Values are considered statistically significant when the *p*-value < 0.05.

		Mean (IQR ^a^)	Beta (95% CI)	*p*
Baseline absolute neutrophil count (ANC, 10^9^/L) (*n* = 3199)				
	Chinese	4.23 (3.15–5.05)	1.00 (Reference)	
	Malay	4.58 (3.38–5.52)	0.12 (−0.05–0.31)	0.18
	Indian	5.00 (3.83–5.89)	0.51 (0.29–0.73)	<0.001
Maximum ANC decrease (10^9^/L) (*n* = 3076)				
	Chinese	2.50 (1.34–3.41)	1.00 (Reference)	
	Malay	2.77 (1.62–3.82)	0.15 (−0.05–0.37)	0.13
	Indian	3.03 (1.77–4.07)	0.41 (0.16–0.67)	0.001

^a^ IQR = Inter-quartile range.

## Data Availability

SGBCC data is available to interested researchers upon request, with certain access limitations. All inquiries should be sent to the scientific steering committee of the Singapore Breast Cancer Cohort (SGBCC). For further information, please email the Principal Investigator, Mikael Hartman, at ephbamh@nus.edu.sg. A list of accessible data is available at https://blog.nus.edu.sg/sgbcc/for–researchers/ (accessed on 1 February 2023). The Joint Breast Cancer Registry dataset may be requested at: https://www.nccs.com.sg/research–innovation/data–on–request (accessed on 1 February 2023).

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
