# Peer review of "How Asian Breast Cancer Patients Experience Unequal Incidence of Chemotherapy Side Effects: A Look at Ethnic Disparities in Febrile Neutropenia Rates"

_cancers, 2023, doi:10.3390/cancers15143590_

Round 1

Reviewer 1 Report

I did read the paper and have a few comments in case they might be helpful. Overall, the manuscript is well-written and the English is good. However, the methods are not always clear. BMI is adjusted for in some models but not others. There is a statement that there is a large amount of missing data for BMI. How was that missingness handled? The tables and figures in the results section need to be more clearly titled without reference to analytic methods or results. In Table 2, they should spell out the variables in the column headings and the in the 'patient characteristics' column. They should clarify the units of Baseline BMI, max BMI drop, and Baseline ANC. Table 3 is very confusing. Why are there no Hazard ratios for the different categories in the 'Chinese' column? Although efforts are made to adjust for several different treatment and demographic factors associated with FN, it is likely that there is still significant residual confounding in the results, as the different races are diagnosed at different stages and treated with different regimens, and there is little information on comorbidities and missing data on details of the treatment regimens. It is unclear how the results of this analysis could lead to treatment improvements in the clinical setting. Overall, this reviewer rates this paper as 'in need of major revisions'.

Author Response

Dear reviewer,

Thank you for your suggestions. Please refer to the content/ attached document for our response.

Reviewer 1: I did read the paper and have a few comments in case they might be helpful. Overall, the manuscript is well-written and the English is good. However, the methods are not always clear.

  1. BMI is adjusted for in some models but not others. There is a statement that there is a large amount of missing data for BMI. How was that missingness handled?

Our response: In our model describing associations between various characteristics and the occurrence of febrile neutropenia (Table 2), individuals with missing values for BMI were excluded from the univariate analysis. Since baseline BMI was not significant, it was not included in further adjusted models. We have clarified this point in the “Materials & Methods” (page 5, line 186 of tracked changes version):

“Patients with missing values for continuous variables were not included in the univariate analysis. ”

In the linear regression showing associations between ethnicity and ANC (Table 4), we only included patients with complete records of ANC and BMI, which could be biased. In this case, after including the excluded individuals, we found that our results are biased towards the null and have included this point in-text as seen below.

Results (page 13, line 306 of tracked changes version):

“Further investigations using proportions test showed that missingness for BMI across ethnicities was not at random (p<0.001). Sensitivity analysis including patients with missing BMI as a separate category was done. The association between ethnicity and baseline ANC became more significant (BetaMalay [95% CI]: 0.24 [0.09 – 0.38], p=0.001; BetaIndian: 0.65 [0.46 -0.84], p<0.001) (Supplementary Table S4). This more pronounced association was also seen in the magnitude of ANC decrease (BetaMalay: 0.23 [0.07 – 0.40], p=0.004; BetaIndian: 0.50 [0.29 -0.71], p<0.001) (Supplementary Table S4).”

Limitations (page 18, line 418 of tracked changes version):

“The dataset has a sizable proportion of subjects (28%) with missing baseline BMI values. However, excluding patients with missing baseline BMI biased the results for absolute neutrophil count towards the null.”

  1. The tables and figures in the results section need to be more clearly titled without reference to analytic methods or results. In Table 2, they should spell out the variables in the column headings and the in the 'patient characteristics' column. They should clarify the units of Baseline BMI, max BMI drop, and Baseline ANC.

Our response: Titles, abbreviations, and column headings for tables and figures have been edited accordingly. Units of measurement have also been added accordingly.

  1. Table 3 is very confusing. Why are there no Hazard ratios for the different categories in the 'Chinese' column?

Our response: Table 3 displays a multinomial logistic regression, which is used to model nominal outcome variables, in which the log odds of the outcomes are modelled as a linear combination of the predictor variables. In this case, we used a polychronic categorical variable, ethnicity, where “Chinese” was used to compare with two other races. For example, results for “Stage” can be interpreted as follow: the odds of a Malay patient developing stage III breast cancer is 2.3 times that of a Chinese patient. We have added clarification for the interpretation in the Results section (page 12, line 292 of tracked changes version).

  1. Although efforts are made to adjust for several different treatment and demographic factors associated with FN, it is likely that there is still significant residual confounding in the results, as the different races are diagnosed at different stages and treated with different regimens, and there is little information on comorbidities and missing data on details of the treatment regimens.

Our response: We agree that there were multiple factors that we were unable to adjust for. We have addressed this point in the limitations (page 18, line 424 of tracked changes version):

“Although efforts were made to adjust for several different treatment and demographic factors associated with FN, it is likely that there is still significant residual confounding in the results”

  1. It is unclear how the results of this analysis could lead to treatment improvements in the clinical setting. Overall, this reviewer rates this paper as 'in need of major revisions'.

Our response: We have added the following explanation to discuss the implications of this study on clinical practice (page 18, line 428 of tracked changes version):

“In summary, the results of this study suggest that there are ethnic differences in the risk of chemotherapy–induced FN. The findings present an opportunity to improve the precision of breast cancer treatment. Taking into consideration the sus-ceptibility of certain ethnic groups to adverse effects, prophylactic or alternative treatments can be utilized to avoid these side effects. Moreover, close monitoring for early signs of infection may be emphasized in high-risk ethnic populations.”

Reviewer 2 Report

Dear Authors, thank you for this study

The strenght of this study is: a huge cohort, the topic being clinicaly-relevant and interesting.

Please kindly see my comments below:

Figure 1 - it is difficult to follow why and what patients with neo/adjuvant therapy were excluded (n=14) the figures should be self-explanatory.

MM: please kindly explain if the study got ethical committee approval or it was waived.

line 336: "and the absence of additional treatments" should be explained what is additional treatment

In conclusion section: what would be authors recommendations in clinical practice (if any) in terms of these findings 

Author Response

Dear reviewer,

Thank you for your suggestions. Please refer to the content/ attached document for our response.

Reviewer 2: Dear Authors, thank you for this study. The strength of this study is: a huge cohort, the topic being clinically-relevant and interesting. Please kindly see my comments below:

  1. Figure 1 - it is difficult to follow why and what patients with neo/adjuvant therapy were excluded (n=14) the figures should be self-explanatory.

Our response: In our study, we defined cases as those who developed febrile neutropenia (FN) during or within 30 days post-chemotherapy. Since these patients underwent both neoadjuvant and adjuvant chemotherapy but only developed FN during their adjuvant therapy, we have combined this group with those who had previous chemotherapy.

  1. MM: please kindly explain if the study got ethical committee approval or it was waived.

Our response: This study was approved by the A*STAR Institutional Review Board (2020–005). SGBCC was approved by the National Healthcare Group Domain Specific Review Board (reference number: 2009/00501) and the SingHealth Centralised Institutional Review Board (CIRB Ref: 2019/2246 [2010/632/B]). The Joint Breast Cancer Registry was approved by the SingHealth Centralised Institutional Review Board (CIRB, 2019/2419 and 2016/3010). Information on ethics approval is included in “Materials and Methods” as follows:

SGBCC (page 3, line 105 of tracked changes version):

“SGBCC was approved by the National Healthcare Group Domain Specific Review Board (reference number: 2009/00501) and the SingHealth Centralised Institutional Review Board (CIRB Ref: 2019/2246 [2010/632/B]). Informed consent was obtained from all patients.”

JBCR (page 3, line 118 of tracked changes version):

“JBCR was approved by the SingHealth Centralized Institutional Review Board (CIRB Ref: CIRB2019/2419 and 2016/3010). As the study relies on de–identified registry–based data, the need to obtain informed consent was waived. This study was approved by the A*STAR Institutional Review Board (2020–005).”

  1. line 336: "and the absence of additional treatments" should be explained what is additional treatment

Our response: Additional treatments refer to non-chemotherapy treatments. We have added further explanations in-text accordingly (page 16, line 356 of tracked changes version):

“Other significant factors associated with FN include age at diagnosis, anthracycline–containing regimens, and the absence of non-chemotherapy treatments, such as radiotherapy, herceptin, and endocrine treatment.”

  1. In conclusion section: what would be authors recommendations in clinical practice (if any) in terms of these findings

Our response: We have added the following explanation to discuss the implications of this study on clinical practice (page 18, line 428 of tracked changes version):

“In summary, the results of this study suggest that there are ethnic differences in the risk of chemotherapy–induced FN. The findings present an opportunity to improve the precision of breast cancer treatment. Taking into consideration the sus-ceptibility of certain ethnic groups to adverse effects, prophylactic or alternative treatments can be utilized to avoid these side effects. Moreover, close monitoring for early signs of infection may be emphasized in high-risk ethnic populations.”